# Permafrost, active layer, and meteorological data (2010–2020) at the Mahan Mountain relict permafrost site of Northeastern Qinghai-Tibet Plateau

Tonghua Wu[1,2*], Changwei Xie[1], Xiaofan Zhu[1], Jie Chen[1], Wu Wang[1], Ren Li[1`], Amin Wen[1], Dong Wang[1], Peiqing Lou[1], Chengpeng Shang[1], Yune La[1], Xianhua Wei[1], Xin Ma[1], Yongping Qiao[1], Xiaodong Wu[1], Qiangqiang Pang[1], Guojie Hu[1]

[1] Cryosphere Research Station on the Qinghai-Tibet Plateau, State Key Laboratory of Cryospheric Science, Northwest Institute of Eco-Environment and Resource, Chinese Academy of Sciences, Lanzhou, Gansu 730000, China

[2] Southern Marine Science and Engineering Guangdong Laboratory (Guangzhou), 511458, China

*Correspondence: Tonghua Wu (thuawu@lzb.ac.cn)

**Abstract:** Relict permafrost presents an ideal opportunity to understand the impacts of climatic warming on the ground thermal regime since it is characterized by a mean annual ground temperature close to 0 ℃ and relatively thin permafrost. The long-term and continuous observations of permafrost thermal state and climate background are of great importance to reveal the links between the energy balance on hourly to annual timescales, to evaluate the variations in permafrost thermal state over multiannual periods and to validate the remote sensing dataset. We present 11 years of meteorological and soil data from the Mahan Mountain relict permafrost site on the northeast of the Qinghai-Tibet Plateau. The meteorological data comprise air and land surface temperature, relative humidity, wind speed and direction, shortwave and longwave downwards and upwards radiation, water vapor pressure, and precipitation on a half-hour timescale. The active layer data include daily soil temperature and soil volumetric water content at five different depths. The permafrost data consist of the ground temperature at twenty different depths up to 28.4 m. The high-quality and long-term datasets are expected to serve as accurate forcing data in land surface models and evaluate remote-sensing products for a broader geoscientific community. The datasets are available from the National Tibetan Plateau/Third Pole Environment Data Center (https://doi.org/10.11888/Cryos.tpdc.271838, Wu and Xie, 2021).

**1 Introduction**

Permafrost is defined as ground that remains at or below 0 ℃ for at least two consecutive years (Van Everdingen, 1998). As a major component of the cryosphere, the area underlain by permafrost ranges from $12.21 \times 10^6 \, \mathrm{km}^2$ to $16.98 \times 10^6 \, \mathrm{km}^2$, or from 12.8% to 17.8% of the terrestrial landscape in the Northern Hemisphere (Zhang et al., 2000). The active layer, which is the top layer of the ground subject to annual thawing and freezing in areas underlain by permafrost, plays an important role in cold regions because most ecological, hydrological, biogeochemical, and pedogenic activities take place within it (Hinzman et al., 1991; Kane et al., 1991; Nelson et al., 2000). The thermal state of permafrost is sensitive to climatic warming. There are increasing evidences indicate that permafrost is warming at both global and regional

scales (Harris et al., 2003; Cheng and Wu, 2007; Romanovsky et al., 2010; Zhao et al.,
2010; Hjort et al., 2018; Biskaborn et al., 2019). Generally, the evidence of permafrost
degradation includes rising mean annual ground temperature, deepening active layer
thickness, talik and thermokarst development, and decreasing permafrost extent
(Cheng and Wu, 2007). Permafrost degradation affects local hydrology, ecosystems,
infrastructure stability, and even feedbacks to the climate system (Nauta et al., 2015;
Walvoord and Kurylyk, 2016; Hjort et al., 2018; Wang et al., 2021; Zhang et al., 2021;
Shogren et al., 2022).
Relict permafrost is usually characterized by high-temperature sporadic
permafrost, where the mean annual ground temperature of permafrost is close to 0 °C.
The relict permafrost presents a favourable opportunity to compare the impacts of
climatic warming on the permafrost and the seasonal frozen ground, as they have
similar climate conditions (Mu et al., 2017). In addition, the different impacts of
vegetation, terrain, and organic matter on the ground thermal regime could be
determined in the relict permafrost regions (Xie et al., 2013). Long-term and
continuous observations of meteorological variables, active layer, and permafrost are
of great importance to understanding the impacts of climatic changes on the ground
thermal regime. It is critical to better understand the energy balance at the ground
surface to enhance our understanding of the heat and moisture exchanges within the
active layer and the permafrost layer. Furthermore, the data on atmospheric conditions
and hydrothermal regimes of the active layer are also of great significance for
validating remote sensing datasets and land surface models in cold regions
(Westermann et al., 2011; Park et al., 2016; Park et al., 2018; Che et al., 2019; Zhao et
al., 2021a). However, on the Qinghai-Tibet Plateau, high-quality and long-term
datasets of meteorological and permafrost data are relatively scarce, especially in the
relict permafrost regions, due to limited logistic support, expensive maintenance costs,
and difficult living environments (Li et al., 2020). It is of great importance to share
the good data for addressing the challenges of climate change and its impacts on
permafrost (Li et al., 2021a). In this paper, the presented data include hourly
meteorological variables, daily soil temperature and soil volumetric water content,

monthly permafrost temperature, and soil physical parameters from the relict permafrost site at the Mahan Mountain.

The Mahan Mountain relict permafrost is located on the northeast Qinghai-Tibet Plateau, which is the peak of the Chinese Loess Plateau and discovered by Li (1986) in fractured bedrock on the Mahan Mountain. It is the only region in the Loess Plateau (China) where permafrost exists. Due to the high mean annual temperature in this region, the permafrost existence can be mainly attributed to two mechanisms. First, the peat layer protects the permafrost from thawing. The organic carbon-rich layer can prevent heating from the air during the warm season as well as the heat loss during the cold season (Du et al., 2012). Second, the high content of ground ice can also favour the presence of the permafrost. It is well known that the phase change of ground ice can absorb a large amount of heat, and thus, the ground temperature will not change significantly in warm permafrost (Biskaborn et al., 2019). In addition, the frequent foggy weather in the area may also decrease the solar radiation and thus favour the presence of permafrost. The characteristics and persistence of the relict high-altitude permafrost on the Mahan Mountain have been demonstrated by Xie et al. (2013).

We present standard meteorological data, including air and land surface temperature, relative humidity, water vapor pressure, wind speed and direction, shortwave downwards and upwards radiation, longwave downwards and upwards radiation, and precipitation. The data cover an 11-year span from January 1, 2010, to December 31, 2020. In addition, field measurements for soil physical parameters at different depths of five sampling sites from October 2015 to August 2016 are also presented, including soil bulk density, soil gravimetric water content, and soil porosity.

**2 Data description**

**2.1 Site description**

The Mahan Mountain relict permafrost observation site (35°44′N and 103°58′E, 3670 m a.s.l.) was established in 2009 by the Cryosphere Research Station on the Qinghai-Tibet Plateau, the Northwest Institute of Eco-Environment and Resource, the

Chinese Academy of Sciences. From 1991 to 1993, Li et al. (1993) drilled 12 boreholes across four transects to evaluate the occurrence of permafrost. Among them, 6 boreholes showed obvious evidence indicating permafrost occurrence. The permafrost mostly emerged in the moist depression regions where vegetation is well developed. The original permafrost area was approximately 0.16 km², the area of which has recently been reduced to 0.13 km² (Xie et al., 2013). The mean annual ground temperature ranges from -0.2 ℃ to -0.3 ℃, which belongs to typical warm permafrost (Cheng and Wu, 2007). The permafrost thickness is approximately 5–40 m, and the active layer thickness ranges from 1.0 m to 1.5 m (Li et al., 1993; Dong et al., 2013; Liu et al., 2015). The existence of an abundant peat layer and ground ice can exert an effective protective effect on the underlying permafrost. Thus, although the permafrost extent is very small, the relict permafrost is not sensitive to climate warming (Xie et al., 2013).

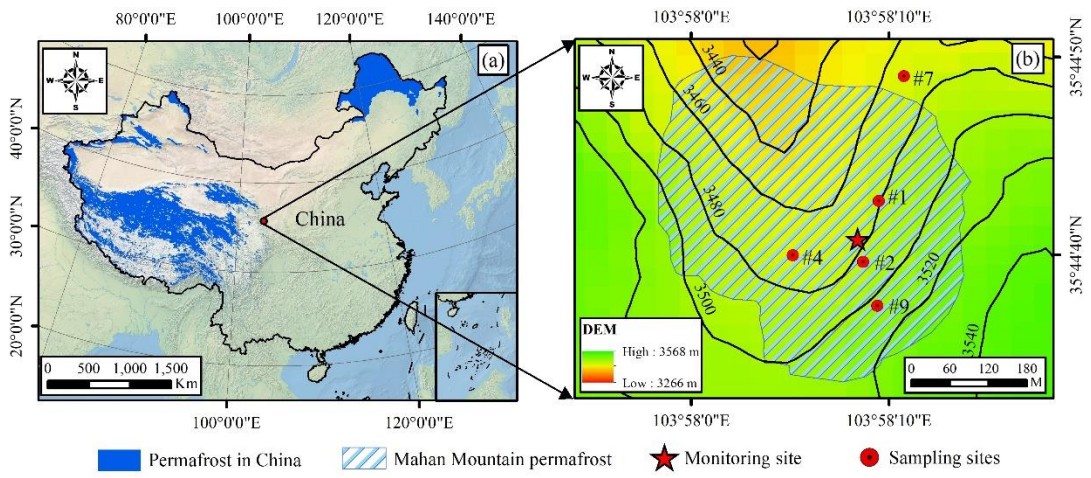

**Figure 1.** The location of Mahan Mountain relict permafrost region in China (a), the spatial distribution of permafrost and monitoring sites in the study region (b). Permafrost distribution data in China are derived from Zou et al. (2017) and Zhang et al. (2019), and the Environmental and Ecological Science Data Center for West China (http://westdc.westgis.ac.cn). The permafrost distribution of the Mahan Mountain is derived from Xie et al. (2013). The high-resolution satellite-derived land cover data are provided by Natural Earth (http://www.naturalearthdata.com).

The climate conditions on the Mahan Mountain are cold and subhumid. The observed mean annual air temperature in the relict permafrost region is approximately -1.4 ℃ from 2010 to 2020, and the duration of negative air temperature exceeds 200

days. The local ground surface is covered by the swamp meadow with approximately 90% coverage. The dominant plant types mainly include *Kobresia humilis*, *K. pygmaea*, and *K. capilifolia* (Sun and Zhao, 1995). Abundant hummocks are well developed and are influenced by high moisture contents and frost heaving effects. A greater ecosystem respiration rate and soil carbon release occurred in the relict permafrost region than in the Arctic permafrost region (Mu et al., 2017).

**2.2 Data description**

The Mahan Mountain meteorological and permafrost observation sites were set up in 2009. The observation details are shown in Fig. 2 and Table 1. There is regular manual maintenance every one or two months, mainly including power system checking, sensor and field cleaning, and data collection. In addition, to prevent the thermistors in the borehole from shifting during the monitoring period, we set up a steel wire running through the borehole, and a cable wrapped with thermistors is fixed to the steel wire, which can ensure that the cable is vertical and prevent the thermistors from moving in the borehole. We also calibrate these thermistors every year at the State Key Laboratory of Frozen Soil Engineering, Chinese Academy of Sciences. However, for the sensors in the active layers, we cannot calibrate the depths of all sensors every year, which may lead to some errors. Moreover, due to the independent power energy from three solar panels, the meteorological data were continuous with high quality.

For the active layer soil temperature and soil water content observations, there were several blank gaps from 2012 to 2014 owing to a broken storage battery. Subsequently, we solved these problems by installing a new storage battery with a larger capacity. Moreover, the permafost borehole suffered water penetration from 2012 to 2016, which caused low-quality permafrost temperature data; we repaired it and manually measured the permafrost temperature at different depths since 2017. The related data introduction is as follows.

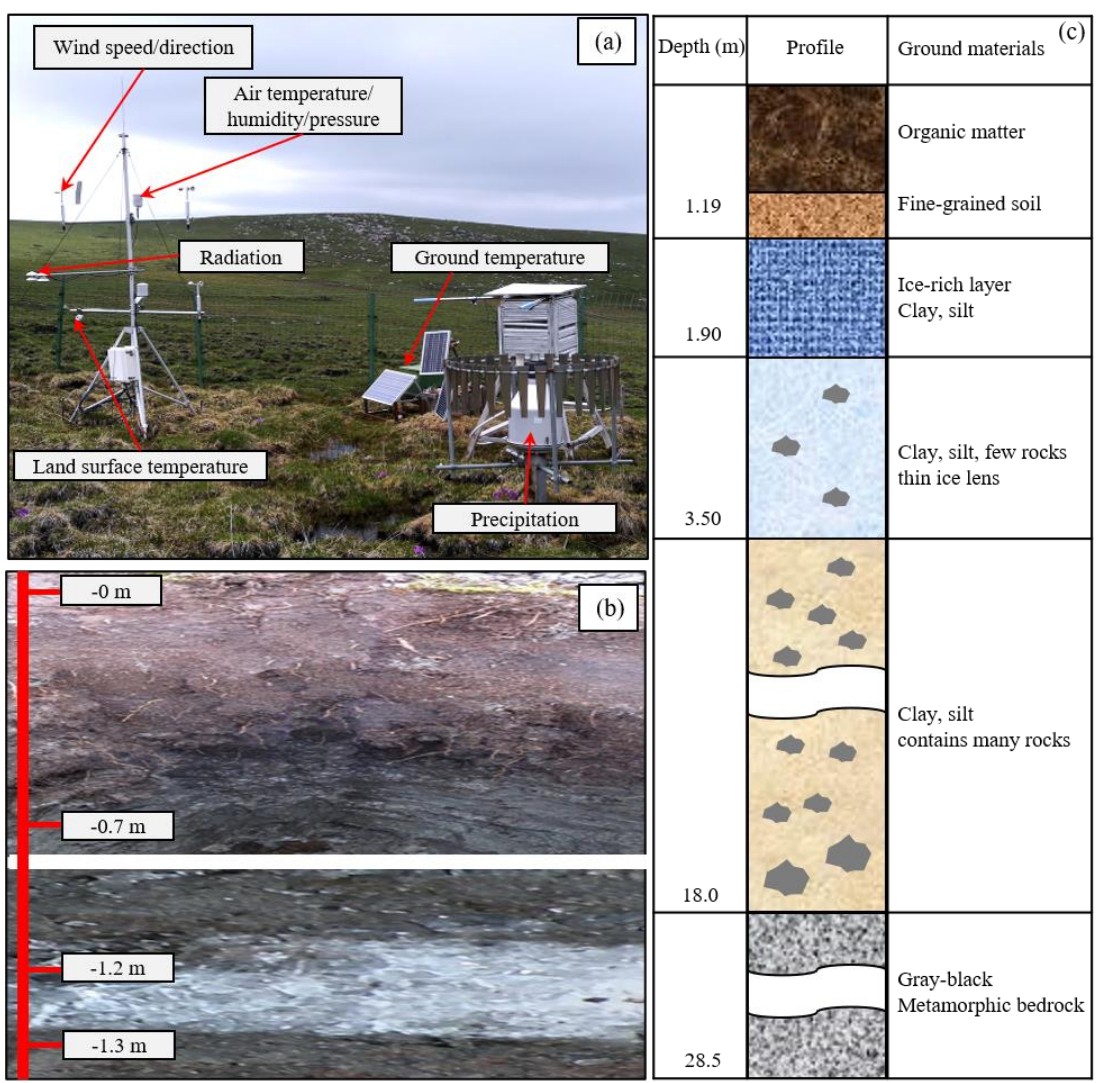

**Figure 2.** The setup of the meteorological and permafrost observation site on the Mahan Mountain. The meteorological monitoring parameters mainly include wind speed and direction, air humidity, radiation, land surface temperature (LST), and precipitation (a); active layer soil profile and ground ice near the permafrost table (b); and soil profile information of the permafrost borehole (c). Figure 2a was recorded on June 15, 2020. Note that we selected flat ground with homogeneous vegetation type to set up the instruments. After some instruments were destroyed by animals, so we set up a fence to protect the instruments. There were slight differences in the vegetation biomass during the following years.

## 2.2.1 Meteorological conditions

The meteorological station of the Mahan Mountain continues to observe a variety of meteorological variables from January 1, 2010, to December 31, 2020 (Table 1). All meteorological variables are monitored in 30-minute intervals (Fig. 2),

and the monitoring data are recorded by a CR1000 data logger (Campbell Scientific,

Inc.). Because the weather observation equipment is regularly maintained, most of the

meteorological data have high quality and continuity with very limited missing data.

The detailed description of each meteorological variable is presented as following

(Table 1).

**Table 1.** List of sensors, accuracy, measuring height, measuring interval, and operation period for

meteorological variables at the Mahan Mountain from January 2010 to December 2020.

| Variable | Sensor | Range | Accuracy | Sensor height | Measuring interval | Unit |
|---|---|---|---|---|---|---|
| Shortwave radiation | CM3, Kipp & Zonen, Netherlands | 0 to 2000 $W/m^2$ | <5% | 2m | 30min | $W/m^2$ |
| Longwave radiation | CM3, Kipp & Zonen, Netherlands | 0 to 2000 $W/m^2$ | <10% | 2m | 30min | $W/m^2$ |
| Air temperature | HMP45C,Vaisala Finland | -40 to 60 ℃ | ±0.2-0.5 ℃ | 2m, 4m | 30min | ℃ |
| Relative humidity | HMP45C,Vaisala Finland | 0 to 100 % RH | ±3% | 2m, 4m | 30min | % |
| Wind speed/direction | 014A, MetOne, USA | 0 to 45 m/s | 0.11m/s | 2m | 30min | $m\ s^{-1}$/ Deg |
| Water vapor pressure | HMP45C,Vaisala Finland | - | ±3% | 2m, 4m | 30min | hPa |
| Precipitation | T200B3 precipitation gauge | 0 to 1000 mm | 0.1% | 1.6m | 1 day | mm |
| Land surface temperature | IRR-P,Vaisala Finland | -55 to 80 ℃ | ±0.3 ℃ | 2m | 30min | ℃ |

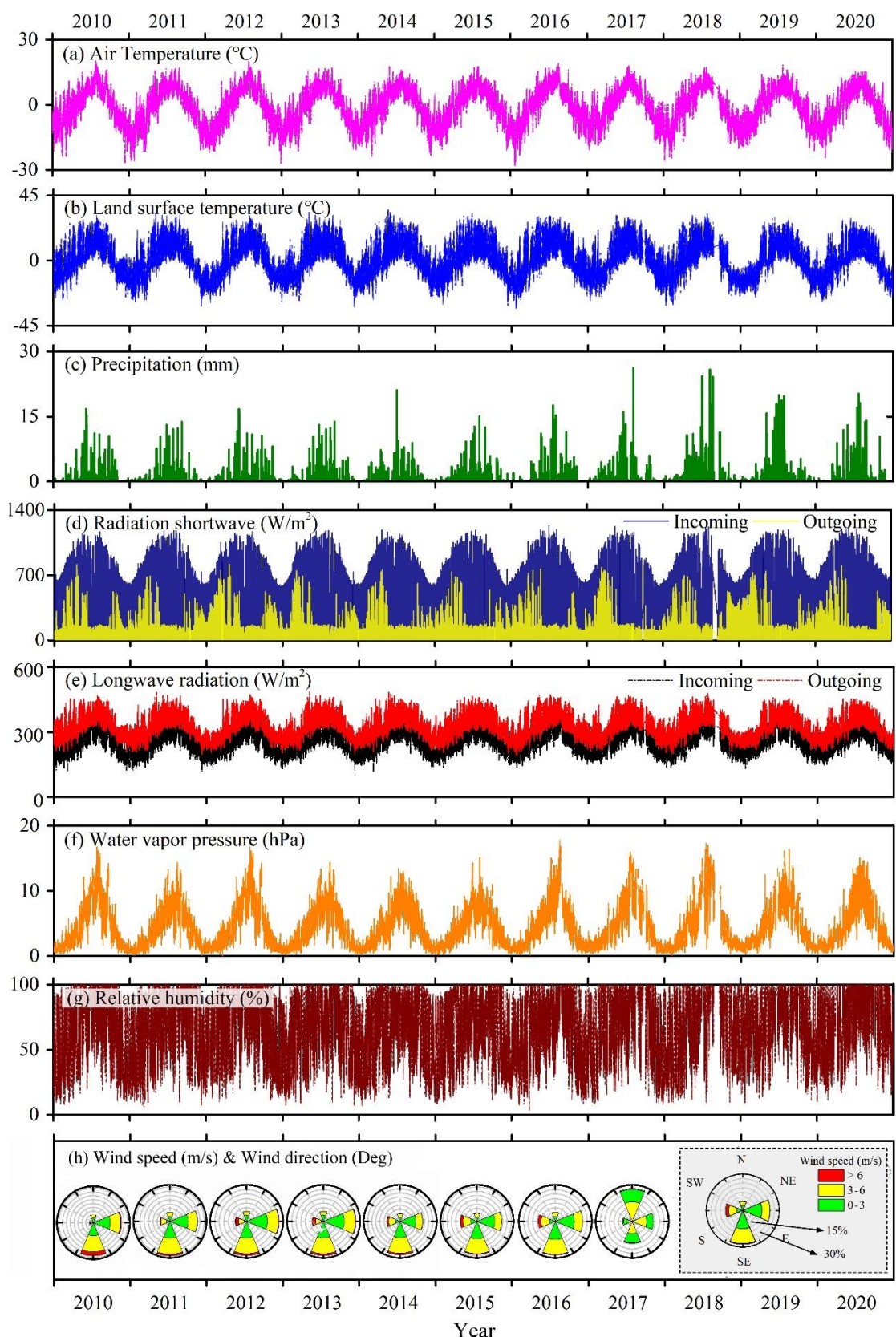

**Figure 3.** Time series of meteorological variables from 2010 to 2020 at the Mahan Mountain, including air temperature at 2 m height (a), land surface temperature (b), precipitation at 1.6 m

height (c), shortwave radiation at 2 m height (d), longwave radiation at 2 m height (e), water vapor pressure at 4 m height (f), relative humidity at 4 m height (g), wind speed & direction at 2 m height (h). The temporal resolution of precipitation data is daily scale, and hourly scale for other all variables.

**Air and land surface temperature**

Air temperature was measured by a shielded HMP45C at heights of 2 m and 4 m above the ground surface. Such sensors were relatively stable, and the data integrity reached to almost 100% with an accuracy of 0.2–0.5 °C. During 2010–2020, the mean annual air temperature at 2 m height ranged from -2.0 °C to -0.7 °C (Fig. 3a). Moreover, the annual variations in air temperature amplitudes were approximately 38.6–47.7 °C (Fig. 3a).

The land surface temperature (LST) was measured by the IRR-P at a height of 2 m above the ground surface through noncontact infrared radiation. At the Mahan Mountain permafrost site, the LST ranged from -33.2 °C to 36.9 °C. The lowest mean annual LST was -2.1 °C in 2012, while the highest mean annual LST was -0.6 °C in 2016, and the 11-year mean LST was -1.4 °C (Fig. 3b).

**Precipitation**

A Geonor T-200B precipitation gauge (1000 mm capacity) was installed at a height of 1.6 m above the ground surface. There is a vibrating-wire sensor within the gauge to measure the total weight of a collection bucket, and a single Alter shield around the gauge can guarantee a higher catch ratio to some extent. In general, the accuracy and sensitivity of this gauge are 0.1% and 0.1 mm, respectively. This gauge has been widely used to as the reference standard in the WMO Solid Precipitation Intercomparison Experiment (WMO-SPICE) (Nitu et al., 2018) and related precipitation intercomparison experiments (Zhao et al., 2021b). Due to the influence of wind disturbance, wetting loss and evaporation loss, some abnormal precipitation values exist. To guarantee the data quality, we have checked related records to decide whether a precipitation event occurred by combining synchronous air temperature and

land surface temperature, shortwave radiation, relative humidity data, and related data were also corrected according to the reference of Domine et al. (2021).

The observed local annual total precipitation was $318.6 \pm 54.3$ mm from 2010 to 2020, and the minimum and maximum annual total precipitation occurred in 2015 and 2018 with values of 258.3 mm and 443.9 mm, respectively (Fig. 3c). In addition, approximately 80% of the annual precipitation is concentrated in the period of May to September, and only no more than 5% of the precipitation occurs in winter.

**Radiation**

Upwards/downwards shortwave and longwave radiations were measured by the Kipp & Zonen CM3 radiometer. The spectral ranges of the shortwave and longwave radiometers are from 0.3 μm to 2.8 μm and from 4.5 μm to 42 μm, respectively. On the Mahan Mountain, the downwards shortwave radiation tended to reach its maximum in spring, followed by summer, and was lowest in winter and autumn. Upwards shortwave radiation also reached its maximum in spring, but the difference was that the downwards shortwave radiation in summer was comparable to that of autumn and winter, or even lower, which was mainly due to the cloudy and rainy weather in summer. The maximum values of upwards/downwards longwave radiation usually occurred in summer, followed by autumn, while the values in winter and spring tended to be lower, which shows similar patterns with the seasonal variations in land surface temperature and air temperature.

**Relative humidity and water vapor pressure**

The relative humidity was measured by shielded HMP45C probes at heights of 2 m and 4 m above the ground surface. However, when in heavy rainfall or fog weather, the observed relative humidity might exceed its physical limits, i.e., 100%. In this case, the relative humidity was corrected to 100% instead (Fig. 3g). The variations in relative humidity were consistent with rainfall events and the variations in air temperature.

The water vapor pressure was calculated from the relative humidity at heights of

2 m and 4 m above the ground surface. Water vapor pressure generally reached its
maximum in summer, followed by autumn, and lowest in spring and winter, which
showed obvious seasonal variations (Fig. 3f).

**Wind speed and wind direction**

The 014A MetOne wind speed and direction sensors were installed at a height of
2 m above the ground surface. The negative values for wind directions were replaced
by 6999. The wind speed and direction during 2010–2017 were continuous with high
quality. Extensive data gaps emerged in the wind direction due to equipment problems
after August 27, 2017. The wind speed data gradually became unavailable after 2019.
The wind speed mainly stayed between 2 m/s and 6 m/s (Fig. 3h).
**2.2.2 Active layer hydrothermal conditions**
**Soil temperature and soil volumetric water content**
The underground soil temperature and soil volumetric water content data in the
active layer were monitored at five depths (10 cm, 30 cm, 80 cm, 100 cm, and 120
cm). The soil temperature were measured by 105T/109 thermistors (Campbell
Scientific, USA) with an accuracy of $\pm 0.1$ ℃. The soil volumetric water content were
measured by the time-domain reflectometry (TDR-100, Campbell Scientific, USA)
with an accuracy of $\pm 0.03$. These sensors were all attached to a CR1000 data logger
(Campbell Scientific, USA) at 30-minute intervals. We finally resampled the
30-minute soil temperature and soil volumetric water content data into daily data by
averaging the half-hourly data within a day.
**Table 2.** List of sensors, accuracy, measuring height and interval, and operation period for soil
temperature and soil volumetric water content within the active layer at the Mahan Mountain
satiation.

| Variable | Sensor | Range | Accuracy | Depth/cm | Measuring interval | Operation period | Unit |
|---|---|---|---|---|---|---|---|

| Soil temperature | 105T, Campbell | -78 to +50 | ±0.1 | 10, 30, 80, 100, 120 | 30min | Jan 2010 – Dec 2020 | ℃ |
| --- | --- | --- | --- | --- | --- | --- | --- |
| Soil volumetric water content | TDR-100, Campbell | 0 to 1 | ±0.03 | 10, 30, 80, 100, 120 | 30min | Jan 2010 – Dec 2020 | $m^3m^{-3}$ |

To obtain highly accurate data, quality control was performed by manually
checking whether there were abnormal or missing data. For the soil temperature data,
the missing data accounted for 17.1% during the period of 2010–2020. The major soil
temperature data gaps were from November 23, 2013, to September 21, 2014. In
addition, we checked the soil temperature data based on the zero-curtain effect,
assuming that the soil properties and water composition did not change during 2010–
2020. For the soil volumetric water content data, the missing and abnormal data
accounted for approximately 30.7% of the entire soil volumetric water content data,
mainly from 2012 to 2014. If the soil volumetric water content data were only missing
in several hours within a day, we interpolated the missing data with the proximity
averaging method. In the case of missing data persisting for a longer time, we filled
them with 6999. Overall, all the missing or abnormal soil temperature and soil
volumetric water content data were replaced with 6999.
According to the soil temperature profile (Fig. 4), the soil temperature in the
active layer shows a seasonal dynamic change. The thawing onset was generally in
the middle of April, and the maximum thawing depth was reached in late September.
The amplitude of the ground temperature in the active layer decreased rapidly with
increasing soil depth. The minimum and maximum values of the soil temperature data
at depths of 10 cm, 30 cm, 80 cm ,100 cm, and 120 cm were -8 ℃ and 9.8 ℃, -6.4 ℃
and 8.4 ℃, -3.1 ℃ and 3.5 ℃, -1.4 ℃ and 1.9 ℃, and -0.74 ℃ and 0.7 ℃,
respectively. The mean annual soil temperature in 2019 reached its maximum during
2010–2020. Under the influence of the freeze–thaw process, the thermal state of the
active layer is not constant during the whole year. In addition, the difference in
thermal conductivity between the frozen and thawed ground causes a "negative
thermal offset", which is defined as the difference between the mean annual soil
temperature at the bottom of the active layer (TTOP) and the mean annual soil surface
(~0 cm) temperature (MAGST) (Burn and Smith, 1988). In this study, the value of
MAGST is larger than +0.97 ℃ (MAST at 10 cm). Therefore, the thermal offset =
TTOP - MAGST= -0.1 ℃ - (> +0.97 ℃) > -1.07 ℃. This result is consistent with the
general understanding of thermal offset in the permafrost regions (Romanovsky and
Osterkamp, 1995).

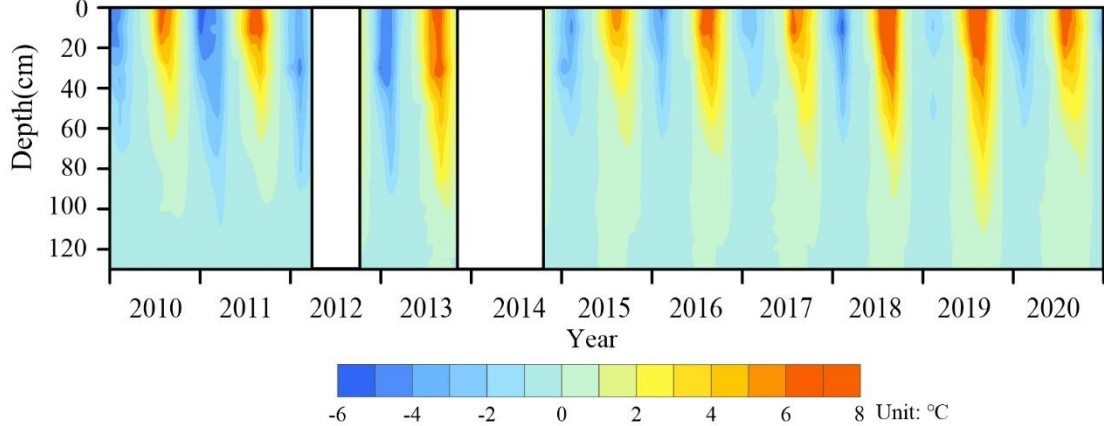

**Figure 4.** Active layer soil temperature profiles during 2010–2020 at the Mahan Mountain
permafrost site. The blank gap stands for the missing data.
As shown in Fig. 5, there were two higher soil volumetric water content zones in
the upper and lower parts of the active layer, which were located at approximately 0–
40 cm and 90–110 cm depths, respectively, and a relatively lower soil volumetric
water content was in the middle part of the active layer. The distribution of abundant
vegetation and peat layers, soil particle fractions, the freeze–thaw process, the ground
ice layer, water channels such as soil pores, and cracks can affect soil water contents.
These factors may account for the abnormal features of soil water contents at the
depths of 40–80 cm and 100 cm (Hincapié and Germann, 2009; Xu et al., 2010; Hu et
al., 2014; Mathias et al., 2015; Zhu et al., 2017). In the thawing season, the soil
volumetric water content reached approximately 0.7 $m^3$ $m^{-3}$ in the upper and lower
parts of the active layer, and was approximately 0.3 $m^3$ $m^{-3}$ to 0.4 $m^3$ $m^{-3}$ in the middle
part of the active layer. In the freezing season, there were significant differences from
the thawing season, and the soil volumetric water content in the middle part of the
active layer was higher than that of the upper and lower parts of the active layer.
Moreover, the soil volumetric water content at 40–90 cm depths exhibited a rapid
increase in the freezing season since 2015, which could reach to 0.4 m$^3$ m$^{-3}$, and the
soil volumetric water content at around 120 cm depth showed a rapid increase in the
freezing season since 2017, with a slightly lower soil volumetric water content than
that of the 40–90 cm depths.

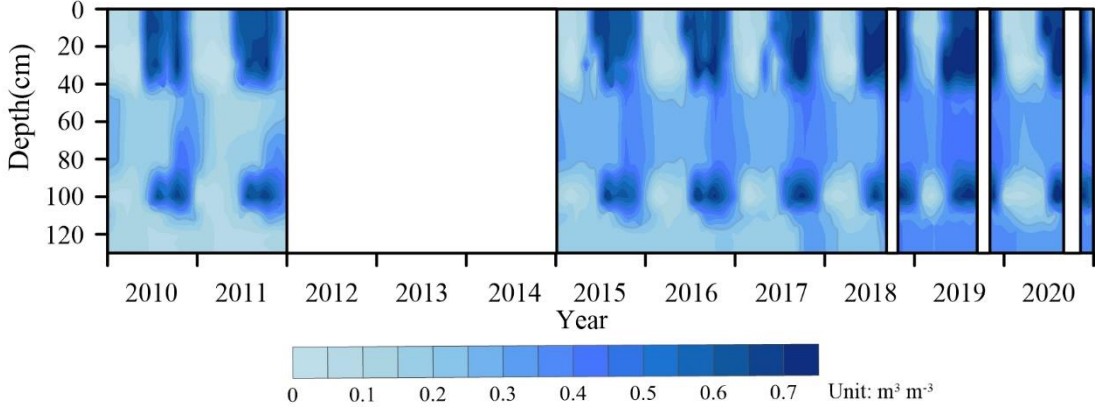


**Figure 5.** Evolution of soil volumetric water content profiles from 2010 to 2020 at the Mahan
Mountain permafrost site. The blank gap stands for the missing data.

The results revealed that the average warming rate of soil temperature at different

depths was 0.056 ℃ /year at the Mahan Mountain from 2010 to 2020 (Fig. 6a). The
highest warming rate of soil temperature was 0.107 ℃ /year at a depth of 30 cm,
while the lowest value was 0.019 ℃ /year at a depth of 120 cm (Fig. 6a). The average
changing trend of the volume soil water content was 0.013 m$^3$ m$^{-3}$/year from 2010 to
2020, and the highest value was 0.026 m$^3$ m$^{-3}$/year at a depth of 120 cm, while the
lowest value was 0.005 m$^3$ m$^{-3}$/year at a depth of 10 cm (Figure 6b).

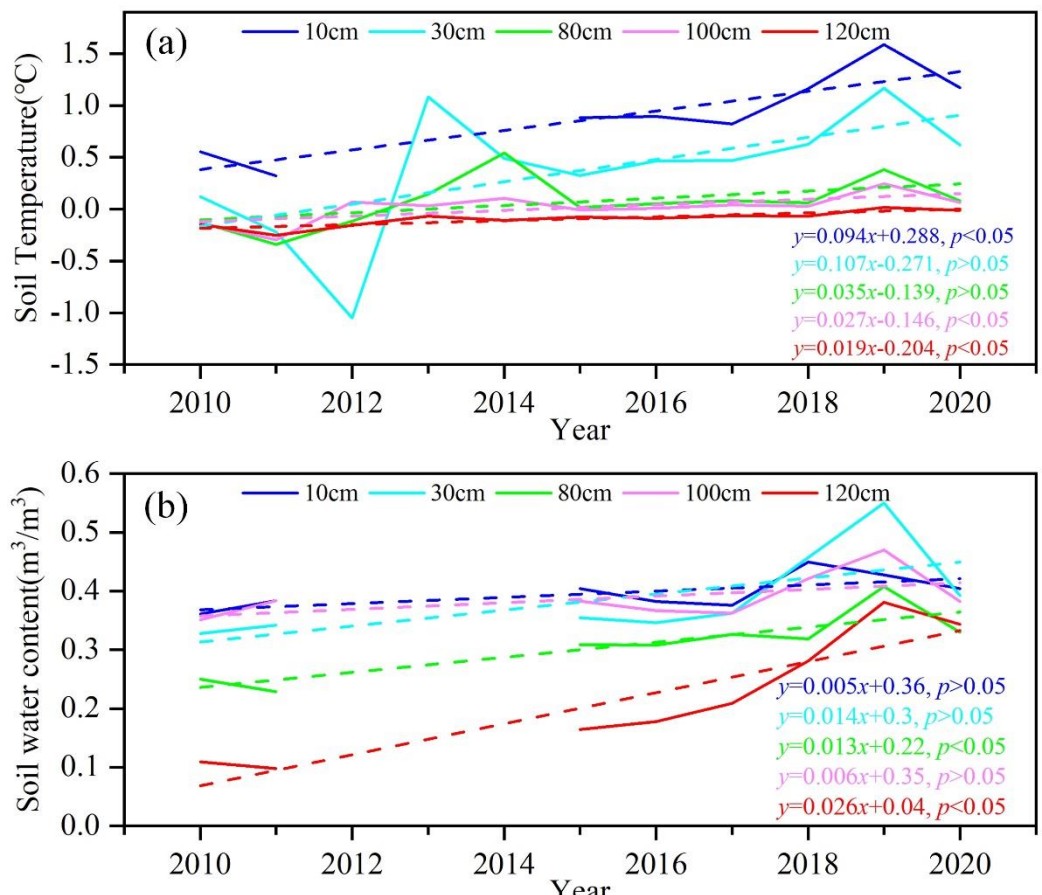

**Figure 6.** Soil temperature and soil volumetric water content at five depths from 2010 to 2020 at the Mahan Mountain permafrost site: soil temperature (a), soil volumetric water content (b).

The active layer thickness (ALT) varied between 107 cm and 150 cm with a mean value of 127 cm from 2010 to 2020 (Fig. 7). The rate of change in ALT was 1.8 cm/year. The increasing rates of ALT in recent decades have varied considerably in different permafrost regions (Table 5).

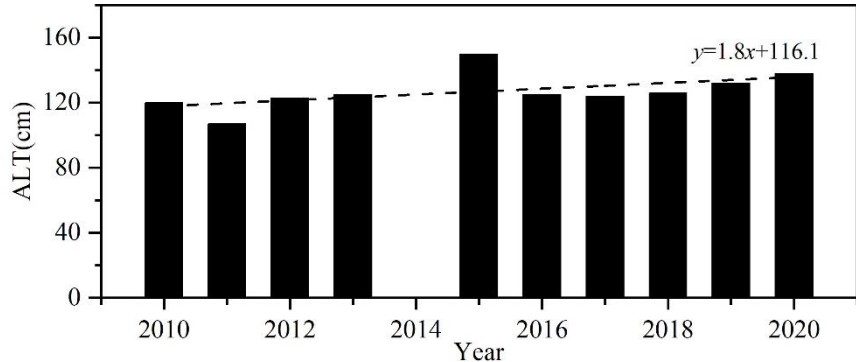

**Figure 7.** The active layer thickness (ALT) from 2010 to 2020 at the Mahan Mountain permafrost

site. The ALT data in 2014 were not available.

## Soil physical parameters

From October 2015 to August 2016, field measurements of soil physical parameter data were carried out by test pit probing and sampling soils, including soil gravimetric water content, soil bulk density, and soil porosity. There were five sampling sites in total. Four sites (#1, #2, #4, and #9) are located in the permafrost region, where the vegetation type is dominated by swamp meadow. Site #7 is located in a seasonally frozen ground region, where the vegetation type is mainly alpine meadow (Table 3). These data can be used as the input parameters in relevant permafrost and land surface process models.

**Table 3** Information on the field sampling site for the soil physical parameters from October 2015 to August 2016 at the Mahan Mountain.

| Sampling site | Elevation (m) | Vegetation type | Frozen soil type |
| --- | --- | --- | --- |
| #1 | 3576.4 | Swamp meadow | Permafrost |
| #2 | 3576.9 | Swamp meadow | Permafrost |
| #4 | 3577.2 | Swamp meadow | Permafrost |
| #7 | 3567.0 | Alpine meadow | Seasonally frozen ground |
| #9 | 3578.7 | Swamp meadow | Permafrost |

Soil samples were obtained in each soil layer using a standard soil sampler (5 cm diameter and 5-cm-high stainless-steel cutting ring). The soil bulk density is estimated using the oven-dry method. Soil porosity is the ratio of nonsolid volume to the total volume of soil, which is calculated by the soil bulk density and specific weight of the soil (Zhao and Sheng, 2015; Indoria et al., 2020). As shown in Table 4, the soil bulk density and soil porosity at sites #4, #7, and #9 presented significant differences at different depths. Site #7, which is located in the seasonally frozen ground, shows a larger soil bulk density ranging from 0.66 $g/cm^3$ to 1.27 $g/cm^3$. As the soil depth increased, the soil bulk density of sites #4 and #7 increased. Soil porosity also showed obvious differences among the three sites, whereas the shallow soil layers exhibited

greater porosity than the deep soil layers. The soil porosity of site #4 ranges from 69.7%
to 85.5%, where the maximum values are found at depths of 0–40 cm.
**Table 4** Soil bulk density and soil porosity within the active layer at different depths from October
2015 to August 2016 at the Mahan Mountain. The location and information of the sampling sites
are shown in Figure 1(b) and Table 3, respectively.

| Depth (cm) | Soil bulk density (g/cm$^3$) | | | Soil porosity (%) | | |
|---|---|---|---|---|---|---|
| | #4 | #7 | #9 | #4 | #7 | #9 |
| 0–10 | 0.34 | 0.66 | 0.45 | 85.5 | 74.3 | 81.0 |
| 10–20 | 0.56 | 0.92 | 0.53 | 76.0 | 65.4 | 76.9 |
| 20–30 | 0.41 | 0.84 | 0.55 | 81.1 | 68.4 | 75.6 |
| 30–40 | 0.37 | 1.03 | 0.56 | 84.5 | 61.8 | 74.5 |
| 40–50 | 0.67 | 1.27 | 0.43 | 74.7 | 53.4 | 76.3 |
| 50–60 | 0.82 | null | 0.46 | 69.7 | null | 83.9 |
| 60–70 | 0.62 | null | 0.45 | 77.1 | null | 83.1 |

Note: "null" stands for no samples.
Moreover, the gravimetric soil water content (GWC) was measured by using the
oven drying method (Zhao and Sheng, 2015). The GWC is the ratio between the
absolute weights of wet and dry soil samples, which can be measured after drying for
24 h at 105 ℃. The GWC at the five sites showed similar profile features. Overall, the
GWC gradually decreased with increasing soil depth (Fig. 8). The GWC at the four
permafrost sites (#1, #2, #4, and #9) shows similar patterns in depth, with their values
ranging from 15% to 250%. The GWC at the seasonal frozen ground site (#7) is only
18.5–77.4%, which is smaller than that at the four permafrost sites (Fig. 8d). In
addition, GWC also presents some monthly differences, such as larger values tending
to occur in June and July in the 10–40 cm layers, which may be caused by abundant
precipitation and thawing processes during this period. The abnormally high value at
a depth of 60 cm at site #9 during August 2016 is likely related to the existence of
subsurface flow.

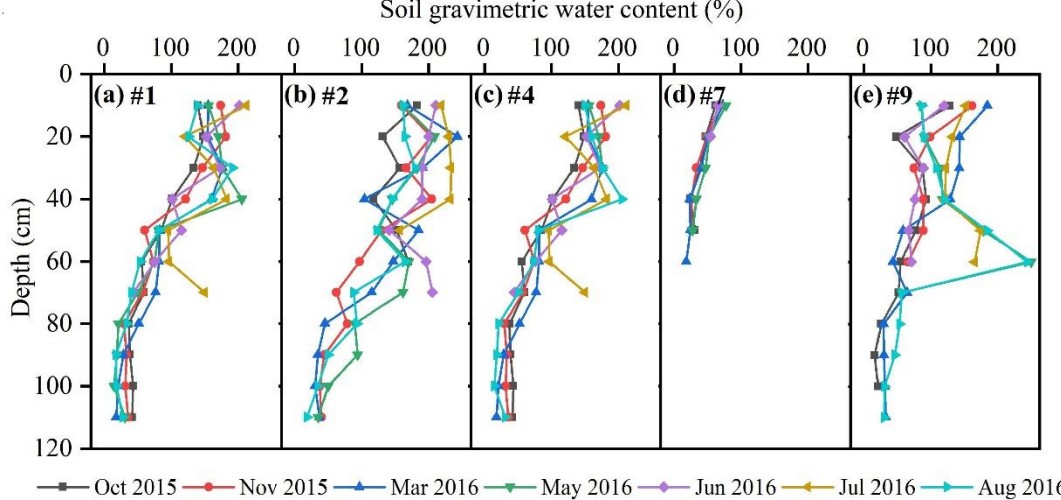


**Figure 8.** Soil gravimetric water content at five sampling sites (#1, #2, #4, #7, and #9) from October 2015 to August 2016 at the Mahan Mountain permafrost region. The location and information of the sampling sites are shown in Figure 1(b) and Table 3, respectively.

### 2.2.3 Permafrost temperature

In August 2008, a borehole with a depth of 28.5 m was drilled to monitor the permafrost temperature. In mid-December 2008, twenty thermistors were installed at different depths in the borehole (0.1 m, 0.4 m, 0.9 m, 1.4 m, 1.9 m, 2.4 m, 3.4 m, 6.4 m, 7.4 m, 9.4 m, 11.4 m, 13.4 m, 15.4 m, 17.4 m, 19.4 m, 21.4 m, 23.4 m, 25.4 m, 27.4 m, and 28.4 m). Thermistor probes made by the Chinese State Key Laboratory of Frozen Soil Engineering at Lanzhou were used to measure the ground temperature. These thermistor probes have a sensitivity of $\pm$ 0.05 ℃ in the lab (Cheng and Wu, 2007). From May 2009, permafrost temperature data for each half-hour were automatically recorded by the datalogger (CR1000, Campbell Scientific, USA). No data were recorded from 2012 to 2016 due to water penetration into the borehole. Since 2017, a digital multimeter has been used to manually measure the permafrost temperature at 13 layers (3 m, 4 m, 5 m, 7 m, 9 m, 11 m, 13 m, 15 m, 17 m, 19 m, 21 m, 23 m, and 25 m) for 2–4 times each month. Quality control was carried out to check whether the data were missing or invalid, which was replaced by 6999 as no data. The ground temperature is then resampled to monthly data.

The records showed that the permafrost temperature at all depths below 2 m was

mostly negative all year round. The location of the permafrost base at this site
exceeded the drilling depth (28.5 m). The soil temperature in the permafrost layer
shows minimum values of approximately -0.2 °C at depths of 10 m to 16 m, close to
-0.1 °C at depths of -2.4 m to -27.4 m, and increased upwards and downwards with a
temperature gradient of ± 0.01 °C/m (Fig. 9). The permafrost temperature data were
not available during 2012–2016 due to the sensor failure. After 2017, a digital
multimeter was used to manually measure the permafrost temperature for 2–4 times
each month. We calculated the annual average permafrost temperature at depths of 9
m and 15 m. The result shows that the annual mean ground temperature at these
depths only showed slight changes during 2010–2020 (Fig. 10).

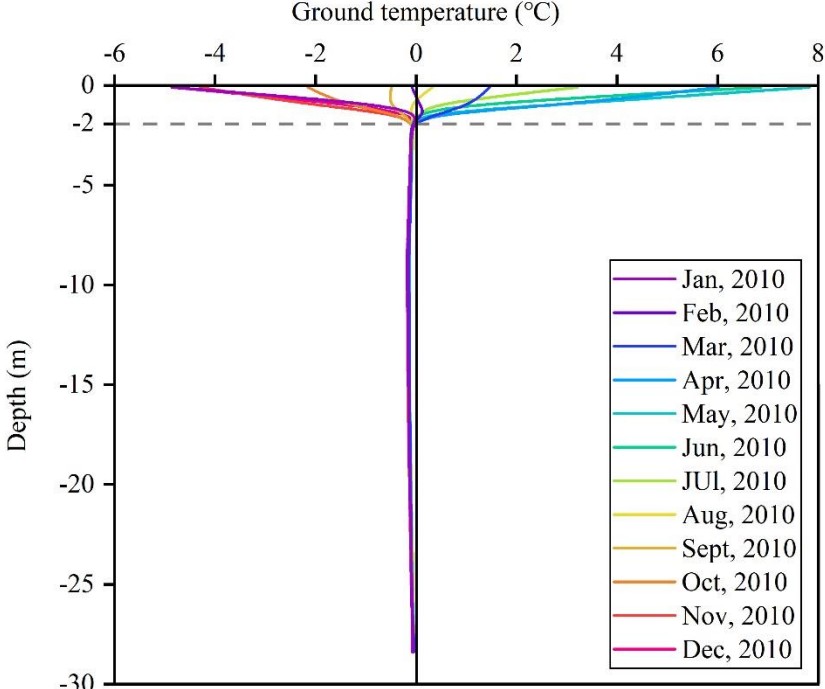


**Figure 9.** Ground temperature in the permafrost borehole drilled in 2010 at the Mahan
Mountain.

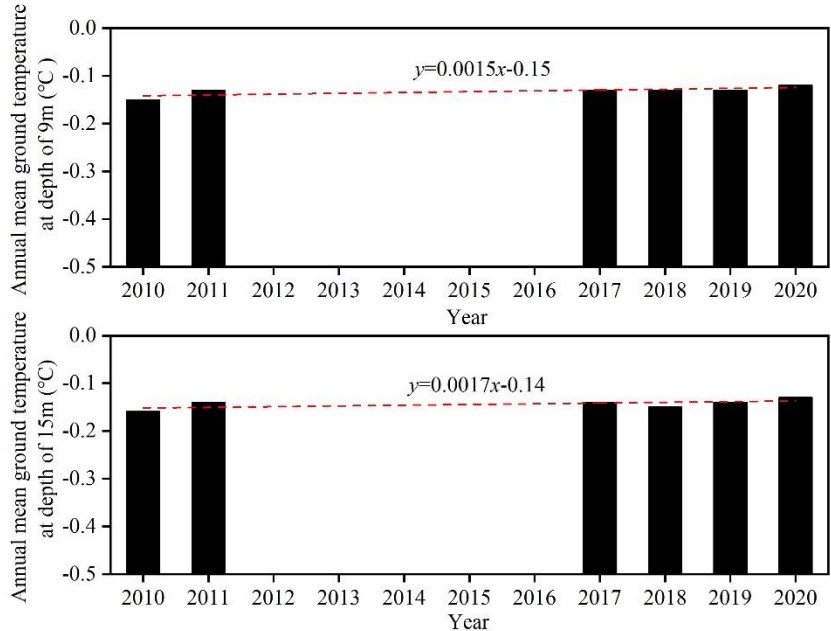

**Figure 10.** The annual mean ground temperature at depths of 9 m and 15 m during 2010–2020 at the permafrost site.

## 2.2.4 Comparison of the variation in permafrost characteristics with other regions

There was an obvious regional difference in the variation in the ALT (Table 5). The change rate of ALT since the 1990s was less than 1 cm/year in the permafrost regions of Alaska, northeastern Siberia, and Antarctica, especially in the permafrost in Canada, which was close to 0 cm/year (Smith et al., 2022). The trends in the permafrost regions of Nordic, northern Russian European, western and central Siberia, and the Qinghai-Tibet Plateau are closer to the results of this paper (Zhao et al., 2019; Smith et al., 2022). The ALT showed the greatest change in permafrost regions of the Swiss Alps (Table 5). In addition, the permafrost temperature change on the Mahan Mountain is significantly lower than that of other regions, which usually have a warming rate greater than 0.15 ℃/decade (Table 5). This pattern can be explained by the existence of a high content of ground ice. The phase change of ground ice can absorb a large amount of heat, and thus, the ground temperature will not change significantly in warm permafrost (Nelson et al., 2001; Biskaborn et al., 2019; Ding et

al., 2019). Moreover, the changes in ALT and permafrost temperature varied greatly from different permafrost regions due to the impact of multiple local factors, such as snow cover, slope aspect, vegetation cover, and soil properties (Ding et al.,2019; Smith et al., 2022). It is worth noting that the different study periods, and variability and continuity of the observed data also have an effect on the results.

**Table 5** Comparison of the change rates of active layer thickness (ALT) and permafrost temperature in different permafrost regions.

| Variable | Area | Variation rate | Study period | Reference |
|---|---|---|---|---|
| Active layer thickness | Alaska North Slope | 0.2 cm/year | 1990–2020 | Smith et al., 2022 |
| | Alaska interior | 0.9 /year | 1990–2020 | |
| | Canada | 0.0 /year | 1991–2018 | |
| | Nordic (including Svalbard and Greenland) | 1.3 cm/year | 1990–2020 | |
| | northern Russian European, western and central Siberia | 1.3 cm/year | 1993–2020 | |
| | northeastern Siberia (including Chuktoka and Kamchatka) | 0.5 cm/year | 1994–2020 | |
| | Swiss Alps | 10.5 cm/year | 1990–2018 | |
| | Antarctica | 0.1 cm/year | 1999–2019 | |
| | Qinghai-Tibet Plateau | 2.17 cm/year | 2004–2018 | Zhao et al., 2019 |
| | Mahan Mountain | 1.8 cm/year | 2010–2020 | This study |
| Permafrost temperature | Arctic continuous permafrost | $0.39\pm0.15$ °C/decade | 2008–2016 | Biskaborn et al., 2019 |
| | Arctic discontinuous permafrost | $0.20\pm0.10$ °C/decade | 2008–2016 | |
| | Mountain permafrost | $0.19\pm0.05$ °C/decade | 2008–2016 | |
| | Antarctica permafrost | $0.37\pm0.10$ °C/decade | 2008–2016 | |
| | Qinghai-Tibet Plateau | 0.15 °C/decade | 2005–2017 | Cheng et al., 2019 |
| | Mahan Mountain | 0.02 °C/decade | 2010–2020 | This study |

## 3 Data availability

The dataset has been available and can be freely download from the National Tibetan Plateau/Third Pole Environment Data Center

(https://data.tpdc.ac.cn/en/disallow/c0a65170-d7cc-4a10-b3fd-39f813cd1387/,

https://doi.org/10.11888/Cryos.tpdc.271838, Wu and Xie, 2021).

**4. Conclusions**

The Mahan Mountain is a relict permafrost site on the northeast of the Qinghai-Tibet Plateau where meteorological and active layer hydrothermal data are automatically acquired and the ground temperature data are manually recorded. This site is dedicated to studies of atmosphere–ground surface interactions and permafrost changes. An 11-year time series of meteorological, active layer and permafrost data is provided. These high-quality and long-term observation data can be used for model validation, including permafrost models, e.g., the CryoGRID 3 model (Westermann et al. 2016), and land surface models, e.g., CLM5 and Noah (Li et al. 2021b). The objective of releasing these data is to improve and validate the permafrost models and land surface models, which face great difficulties in modelling mountain permafrost dynamics.

**Author contributions**

Tonghua Wu designed the research and obtained funding. Changwei Xie and Wu Wang deployed and maintained the instruments. Xiaofan Zhu, Jie Chen, Amin Wen, Dong Wang, Peiqing Lou, Chengpeng Shang, Yune La, Xianhua Wei, Xin Ma and Yongping Qiao analyzed the data and prepared the data files. Ren Li, Xiaodong Wu, and Guojie Hu conducted the field work. Tonghua Wu wrote the paper with inputs from the co-authors and coordinated the analysis and contributions from all co-authors.

**Competing interests**

The authors declare that they have no conflict of interest.

**Acknowledgements**

This work was financially supported by the CAS "Light of West China" Program,the National Natural Science Foundations of China (41771076, 41690142, 41961144021). We thank in particular Professor Lin Zhao from Nanjing University of

Information Science & Technology for his long-term support to maintain the

observation.

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
