# Peer review of "Permafrost, active layer, and meteorological data (2010– 2020) at the Mahan Mountain relict permafrost site of Northeastern Qinghai-Tibet Plateau"

_Earth System Science Data, 2021_

## Author Comment (AC1)

Dear Editor and the reviewer,

We are very grateful to your constructive comments and thoughtful suggestions. Based on these comments and suggestions, we have made thorough revisions to the original manuscript. In addition, we polished our language by the highly qualified native English-speaking editors at AJE in the revised manuscript. The changes made to the text are highlighted in blue so that they may be easily identified in the revised manuscript. Above these have led to an improvement of the paper, and we hope the revised manuscript is suitable for publication in the journal.

Yours sincerely,

Tonghua Wu on behalf of all co-authors
* * *
**Response to comments:**

The paper by Wu et al., presents 11 years of meteorological and soil data in a relict permafrost site of the Mahan Mountain on the northeast of the QTP. The paper is generally well organized and clear to me. As a permafrost researcher, I appreciate the considerable efforts taken by the authors to the permafrost community. I also very much welcome the publication of valuable permafrost datasets.

**Major comment**

1. **GST vs LSM**: "The ground surface temperature (GST) was measured by the IRP-P at a height of 2 m above the ground surface through non-contact infrared radiation" (P10, L167–168). In this context, the measured value is the radiative skin temperature of the land surface. The surface could be snow, grass, and a mixture of them, right? I would keep the "GST" for soil temperature and "LSM" for ground skin temperature. I hence suggest changing the GST to LSM throughout the manuscript.

   **Response**:

   Thanks for the suggestion. We totally agree with your suggestions. In

the revised manuscript, we changed GST to LST in the context.

2. **Permafrost and Active Layer**: Based on the authors' title—"Permafrost, active layer and meteorological data (2010–2020) from a relict permafrost site at Mahan Mountain", I would expect the detailed info of permafrost & active layer would be present. However, the paper in its current format is somehow unfocused, and the permafrost/active layer is very lightly discussed and seems equal or even less important than the meteorological data. Hence, I suggest enhancing the statistics of permafrost and active layer conditions and changes during 2010–2020. Since authors have a 11-years' time-series, this could be easily done by

(1) adding permafrost temperature profile (MAGT) in different years info Figure 7, so that readers could see permafrost temperature changes;

**Response**:

Thanks for your suggestion. The permafrost temperature data were not available during 2012-2016 due to the sensor failure. After 2017, a digital multimeter was used to manually measure the permafrost temperature for 2–4 times each month. Therefore, only 6 years of data are available. In order to analyze the changes of the permafrost temperature, we calculated the annual average permafrost temperature at the depth of 9 m and 15 m. The result shows that the annual mean ground temperature at these depths only showed slight changes during 2010–2020. To be clear, we explained this and added the figure in the result section as follows:

The permafrost temperature data were not available during 2012–2016 due to the sensor failure. After 2017, a digital multimeter was used to manually measure the permafrost temperature for 2–4 times each month. We calculated the annual average permafrost temperature at depths of 9 m and 15 m. The result shows that the annual mean ground temperature at these depths only showed slight changes during 2010–2020 (Fig. 10) (Line 386-391).

[Figure]

**Figure 10.** The annual mean ground temperature at depths of 9 m and 15 m during 2010–2020 at the permafrost site.

(2) presenting active layer thickness (conditions and changes) based on soil temperature;

**Response**:

Thanks for the suggestion. In the revised manuscript, we added the temporal changed in active layer hydro-thermal conditions and its thickness, which are as follows:

The results revealed that the average warming rate of soil temperature at different depths was 0.056 °C /year at Mahan Mountain from 2010 to 2020 (Fig. 6a). The highest warming rate of soil temperature was 0.107 °C /year at a depth of 30 cm, while the lowest value was 0.019 °C /year at a depth of 120 cm (Fig. 6a). The average changing trend of the volume soil water content was 0.013 $m^3$ $m^{-3}$/year from 2010 to 2020, and the highest value was 0.026 $m^3$ $m^{-3}$/year at a depth of 120 cm, while the lowest value was 0.005 $m^3$ $m^{-3}$/year at a depth of 10 cm (Figure 6b) (Line 305-311).

[Figure]

$y=0.094x+0.288, p<0.05$
$y=0.107x-0.271, p>0.05$
$y=0.035x-0.139, p>0.05$
$y=0.027x-0.146, p<0.05$
$y=0.019x-0.204, p<0.05$

$y=0.005x+0.36, p>0.05$
$y=0.014x+0.3, p>0.05$
$y=0.013x+0.22, p<0.05$
$y=0.006x+0.35, p>0.05$
$y=0.026x+0.04, p<0.05$

**Figure 6.** Soil temperature and soil volumetric water content at five depths from 2010 to 2020 at Mahan Mountain permafrost site: soil temperature (a), soil volumetric water content (b).

The active layer thickness (ALT) varied between 107 cm and 150 cm with a mean value of 127 cm from 2010 to 2020 (Fig. 7). The rate of change in ALT was 1.8 cm/year. The increasing rates of ALT in recent decades have varied considerably in different permafrost regions (Table 5) (Line 315-318).

[Figure]

$y=1.8x+116.1$

**Figure 7.** The active layer thickness (ALT) from 2010 to 2020 at Mahan Mountain permafrost site. The ALT data in 2014 were not available.

(3) discussing why permafrost could be relict here. This is the most unique feature for this site. The permafrost temperature is very very close to 0°C (i.e., around -0.1°C, and only slightly increased. With the presence of massive ground ice at this site, air temperature warming would mostly lead to significant phase change rather than temperature increase. Also, the thick peat layer and cloudy/foggy weather in summer are favorable for the presence of permafrost.

**Response**:

Thanks for the suggestion. The Mahan Mountain is the only region in the Loess Plateau (China) where permafrost exists. The permafrost is typical warm permafrost and remains in a very fragile and sensitive state. There is an important protective effect from the peat layer and ground ice. In addition, the foggy weather in summer is very frequent, which can shield local solar radiation into soil layer to some extent. Therefore, local permafrost could be relict here. In the revised manuscript, we explained this in the discussion section as follows:

The Mahan Mountain is the only region in the Loess Plateau (China) where permafrost exists. Due to the high mean annual temperature in this region, the permafrost existence can be mainly attributed to two mechanisms. First, the peat layer protects the permafrost from thawing. The organic carbon-rich layer can prevent heating from the air during the warm season as well as the heat loss during the cold season (Du et al., 2012). Second, the high content of ground ice can also favour the presence of the permafrost. It is well known that the phase change of ground ice can absorb a large amount of heat, and thus, the ground temperature will not change significantly in warm permafrost (Biskaborn et al., 2019). In addition, the frequent foggy weather in the area may also decrease the solar radiation and thus favour the presence of permafrost. (Line 79-88)

**References:**

Biskaborn, B. K., Smith, S. L., Noetzli, J., Matthes, H., Vieira, G., Streletskiy, D. A., Schoeneich, P., Romanovsky, V. E., Lewkowicz, A. G., Abramov, A.,

Allard, M., Boike, J., Cable, W. L., Christiansen, H. H., Delaloye, R., Diekmann, B., Drozdov, D., Etzelmüller, B., Grosse, G., Guglielmin, M., Ingeman-Nielsen, T., Isaksen, K., Ishikawa, M., Johansson, M., Johannsson, H., Joo, A., Kaverin, D., Kholodov, A., Konstantinov, P., Kröger, T., Lambiel, C., Lanckman, J.-P., Luo, D., Malkova, G., Meiklejohn, I., Moskalenko, N., Oliva, M., Phillips, M., Ramos, M., Britta, A., Sannel, K., Sergeev, D., Seybold, C., Skryabin, P., Vasiliev, A.,Wu, Q., Yoshikawa, K., Zheleznyak,M., and Lantuit, H.: Permafrost is warming at a global scale, Nat. Commun., 10, 264, https://doi.org/10.1038/s41467-018-08240-4, 2019.

Du, R., Peng, X., Frauenfeld, O.W, Sun, W., Liang, B., Chen, C., Jin, H., Zhao, Y.; The role of peat on permafrost thaw based on field observations, Catena, 208: 105772, https://doi:10.1016/j.catena.2021.105772, 2022.

3. **Language**: The English need to be carefully checked and revised by native speakers.

**Response**:

Thanks for the suggestion. We polished our language by American Journal Experts (https://www.aje.com/) which is a partner of many publishing groups. The changes were highlighted in blue so that they may be easily identified. The editing certificate by AJE were presented as follows:

[Figure]

**Specific comments**

P2, L39: The permafrost extent is from Zhang et al., (2000), right? If so, I would only cite the related reference and remove the others here.

> **Response**:
>
> Yes. It is from the literature of Zhang et al. (2000). In the revised manuscript, we removed other unrelated references (Line 37).

P3, L44: There are "increasing"...

> **Response**:
>
> Changed. (Line 41-42).

P4, L98: This is a repeat of L83.

> **Response**:
>
> Thanks for the suggestion. In the revised manuscript, we have deleted this sentence in the Line of 83-85.

P5, L111: Relict permafrost will not disappear in the next 40–50 years, this means it is not sensitive to climate warming...

> **Response**:
>
> Agree. In the revised version, we changed this sentence as: The relict permafrost is not sensitive to climate warming. (Line 114-115)

P10, L166: was ranging → ranged.

> **Response**:
>
> Changed. (Line 181)

P14, L269--271: Please remove the missing data info in the figure caption as this has already shown well in the Figure. In such a case, the sentence could be much short, i.e. The blank gap stands for the missing data.

> **Response**:
>
> Thanks for the suggestion. In the revised manuscript, we removed the

missing data information and replaced it with "The blank gap stands for the missing data". (Line 283). In addition, we also made changes for similar errors throughout the manuscript.

P18, L350: In general, it is difficult to distinguish the permafrost model and LSM, it largely depends on the research purpose. Permafrost models, i.e., CryoGRID 3 model (Westermann et al., 2016) has the land surface processes (snow, energy bucket), and permafrost physics have also been implemented into the land surface models, i.e., CLM5, Noah, CLASSIC. What about changing to "...valid models..."?

**Response**:

Agree. We rewrote the sentence as "These high-quality and long-term observation data can be used for model validation, including permafrost models, e.g., the CryoGRID 3 model (Westermann et al. 2016), and land surface models, e.g., CLM5 and Noah (Li et al. 2021)." (Line 428-431)

**References:**

Westermann, Sebastian; Langer, Moritz; Boike, Julia; Heikenfeld, Max; Peter, Maria & Etzelmuller, Bernd (2016). Simulating the thermal regime and thaw processes of ice-rich permafrost ground with the land-surface model CryoGrid 3. Geoscientific Model Development. ISSN 1991-959X. 9(2), p. 523–546. doi: 10.5194/gmd-9-523-2016.

Li, X., Wu, T., Wu, X., Chen, J., Zhu, X., Hu, G., Li, R., Qiao, Y., Yang, C., Hao, J. and Ni, J., 2021. Assessing the simulated soil hydrothermal regime of the active layer from the Noah-MP land surface model (v1. 1) in the permafrost regions of the Qinghai–Tibet Plateau. Geoscientific Model Development, 14(3), pp.1753-1771.

**Tables & Figures**

Table 1: The operation period is the same for all measured variables, right? In this case, I would suggest removing the column and putting the temporal coverage info in the table caption.

**Response**:

Thanks, we removed the column of operation period, and added the information in the title of Table 1.

Figure 6 Should the unit of y-axis be "cm"?

**Response**:

Yes, we changed the unit in Figure 6, which is as follows:

[Figure]

**Figure 8.** Soil gravimetric water content at five sampling sites (1#, 2#, 4#, 7#, and 9#) from October 2015 to August 2016 at Mahan Mountain. The location and information of the sampling sites are shown in Figure 1(b) and Table 3, respectively.

Figure 3: Please somehow change the y-axis range of relative humidity (g). A maximum value of 120 is not reasonable here (as you mentioned in L208). A tricky would be to give the y-axis range a little bit greater than 100%, but only show the value labels between 0–100.

**Response**:

In the revised manuscript, we recreated the Figure 3 as follows:

[Figure]

**Figure 3.** Time series of meteorological variables from 2010 to 2020 at Mahan Mountain, including air temperature at 2 m height (a), land surface temperature (b), precipitation at 1.6 m height (c), shortwave radiation at 2 m height (d), longwave radiation at 2 m height (e), water vapor pressure at 4 m height (f), relative humidity at 4 m height (g), wind speed & direction at 2 m height (h). The temporal resolution of precipitation data is daily scale, and hourly scale for other all variables.

---

## Author Comment (AC2)

Dear Editor and the reviewer,

We are very grateful to your constructive comments and thoughtful suggestions. Based on these comments and suggestions, we have made thorough revisions to the original manuscript. In addition, we polished our language by the highly qualified native English-speaking editors at AJE in the revised manuscript. The changes made to the text are highlighted in blue so that they may be easily identified in the revised manuscript. Above these have led to an improvement of the paper, and we hope the revised manuscript is suitable for publication in the journal.

Yours sincerely,

Tonghua Wu on behalf of all co-authors

**Response to comments:**

As a staff focusing on model development and application on frozen ground and cold region hydrology, I have read this manuscript with great interest. Permafrost is an indicator of climate change. The response of permafrost to climate change is one of the critical issues in cryospheric science. The 10-year record of permafrost, active layer and meteorological data presented by Tonghua Wu et al. from a relict permafrost site of Mahan Mountain in the northeast of Qinghai-Tibet Plateau is a valuable dataset for permafrost and climate research community, especially important for model development and validation. This is a clearly written paper and the overall structure of the manuscript is well organized. In my opinion, this manuscript and dataset are an important contribution to permafrost science. Therefore I recommend that the manuscript to be accepted after some minor revisions. And I also look forward to an open access for this dataset as soon as possible after protection period.

Here are my major comments:

1 I strongly suggest to add more detailed temporal variations analysis for active layer

hydro-thermal condition and permafrost temperature, particularly for ALT; in addition, the comparison analysis with other permafrost regions is suggested to add.

**Response**:

Thanks for the suggestion. In the revised manuscript, we have added more detailed temporal variations analysis for active layer hydro-thermal condition and permafrost temperature, and added a section for the comparison analysis with other permafrost regions as follows:

[revised manuscript text omitted]

**References:**

Ding, Y., Zhang, S., Zhao, L., et al., 2019. Global warming weakening the

inherent stability of glaciers and permafrost. Science Bulletin, 64(4): 245-253.

Nelson, F.E., Anisimov, O.A. and Shiklomanov, N.I., 2001. Subsidence risk from thawing permafrost - The threat to man-made structures across regions in the far north can be monitored. Nature, 410(6831): 889-890.

Smith, S.L., O'Neill, H.B., Isaksen, K. et al. 2022. The changing thermal state of permafrost. Nat Rev Earth Environ 3, 10–23.

Zhao, L., Zou, D., Hu, G., et al., 2021. A synthesis dataset of permafrost thermal state for the Qinghai-Tibet (Xizang) Plateau, China. Earth System Science Data, 13(8): 4207-4218.

2 In the introduction section, the authors need to state the particularity of the relict permafrost site at this site, including why permafrost could be relict here.

**Response**:

Thanks for the suggestion. The Mahan Mountain is the only region in the Loess Plateau (China) where permafrost exists. Due to the high mean annual temperature in this region, the permafrost existence can be mainly attributed to two mechanisms. First, the peat layer protects the permafrost from thawing. The organic carbon-rich layer can prevent heating from the air during the warm season as well as the heat loss during the cold season (Du et al., 2012). Second, the high content of ground ice can also favour the presence of the permafrost. It is well known that the phase change of ground ice can absorb a large amount of heat, and thus, the ground temperature will not change significantly in warm permafrost (Biskaborn et al., 2019). In addition, the frequent foggy weather in the area may also decrease the solar radiation and thus favour the presence of permafrost. In the revised manuscript, we added this information in the introduction. (Line 78-88)

3 In Figure 5, some low-values of soil moisture occurred in the depth of 40-80cm, and some high-values occurred near 100cm or so, what's the reason?

**Response**:

As shown in Fig. 5, there were two higher VWC zones in the upper and lower part of active layer, which were located at around 0–40cm and 90–110cm depths, respectively, and a relatively lower VWC was in the middle part of active layer. There were three possible reasons for the situation, which were as follows:

First, the abundant vegetation and peat layer in shallow soil layer can retain moisture from precipitation infiltration and melting of active layer. In addition, the different soil particle fractions at different depths might influence precipitation infiltration process (Grote et al., 2010; Mathias et al., 2015; Zhu et al., 2017).

Second, the freezing process of active layer is bidirectional. Soil water migrates toward two freezing fronts in this process, then the middle of active layer will become dehydrated and with a low soil moisture content. During the thawing process, soil temperature in the active layer is relatively high in the upper and low in bottom layer, and the temperature gradient will drive soil water to migrate downward. As a result, soil moisture in the deep layers will increase significantly, and soil water at the upper layer will also increase (Zhao et al., 2000; Hu et al., 2014).

Third, some pores, cracks or other water channels probably exist at depths of 40 – 80 cm, which could shape the preferential flows patterns in the soil profile (Xu et al., 2010). These factors may also contribute to the high soil water storage below 80 cm (Greco, 2002; Hincapié and Germann, 2009).

In the revised manuscript, we have added this information, which is as follows:

The distribution of abundant vegetation and peat layers, soil particle fractions, the freeze–thaw process, the ground ice layer, and water channels such as soil pores, and cracks can affect soil water contents. These factors may account for the abnormal features of soil water contents at the depths of 40–80 cm and 100 cm (Hincapié and Germann, 2009; Xu et al., 2010; Hu et al., 2014;

Mathias et al., 2015; Zhu et al., 2017) (Line 287-292).

4 I suggest to add data sources of topographic base in Figure 1a, and to add unit for

DEM in Figure 1b.

**Response**:

Added as follows:

[Figure]

**Figure 1.** Location (a), topographical map and observation site (b) of Mahan Mountain relict

permafrost region. Permafrost distribution data in China are derived from Zou et al. (2017) and

Zhang et al. (2019), and the Environmental and Ecological Science Data Center for West

China (http://westdc.westgis.ac.cn). The permafrost distribution of Mahan Mountain is derived

based on a field survey. The high-resolution satellite-derived land cover map data are provided

by Natural Earth (http://www.naturalearthdata.com).

5 The language should be polished by English-native-speakers before its acceptance

for publication.

**Response**:

Thanks for the suggestion. We polished our language by American Journal

Experts (https://www.aje.com/) which is a partner of many publishing groups.

The changes were highlighted in blue so that they may be easily identified. The

editing certificate by AJE were presented as follows:

[Figure]

**Editing Certificate**

This document certifies that the manuscript

**Permafrost, active layer and meteorological data (2010–2020) from a relict permafrost site at Mahan Mountain, Northeast of Qinghai-Tibet Plateau**

prepared by the authors

**Tonghua Wu, Changwei Xie, Xiaofan Zhu, Jie Chen, Wu Wang, Ren Li, Amin Wen, Dong Wang, Peiqing Lou, Chengpeng Shang, Yune La, Xianhua Wei, Xin Ma, Yongping Qiao, Xiaodong Wu, Qiangqiang Pang, Guojie Hu**

was edited for proper English language, grammar, punctuation, spelling, and overall style
by one or more of the highly qualified native English speaking editors at AJE.

This certificate was issued on **February 8, 2022** and may be verified
on the AJE website using the verification code **C85B-8D21-C762-B3C5-0241** .

[Figure]

Neither the research content nor the authors' intentions were altered in any way during the editing process. Documents receiving this certification should be English-ready for publication; however, the author has the ability to accept or reject our suggestions and changes. To verify the final AJE edited version, please visit our verification page at aje.com/certificate. If you have any questions or concerns about this edited document, please contact AJE at support@aje.com.

AJE provides a range of editing, translation, and manuscript services for researchers and publishers around the world.
For more information about our company, services, and partner discounts, please visit aje.com.

---

## Author Comment (AC3)

Dear Editor and the reviewer,

We are very grateful to your constructive comments and thoughtful suggestions. Based on these comments and suggestions, we have made thorough revisions to the original manuscript. In addition, we polished our language by the highly qualified native English-speaking editors at AJE in the revised manuscript. The changes made to the text are highlighted in blue so that they may be easily identified in the revised manuscript. Above these have led to an improvement of the paper, and we hope the revised manuscript is suitable for publication in the journal.

Yours sincerely,

Tonghua Wu on behalf of all co-authors
* * *
**Response to comments:**

As reviewers 1 and 2 stated, I agree that the dataset would be useful for permafrost communities, including general geocryology, permafrost modeling, etc. This manuscript was generally organized well. The observational system is maintained well during the past decade. Besides the comments from reviewers 1 and 2, I also have some further comments.

Major comments

1 As a data description paper, the authors should make all data descriptions clear and correct. I list a few of the issues here (not limit to these. The authors should read and double-check thoroughly).

**Response**:

We are very grateful to your constructive comments and thoughtful suggestions. Based on these comments and suggestions, we have made thorough revisions to the original manuscript. The changes made to the text are highlighted in blue so that they may be easily identified in the revised manuscript.

We have listed our point-to-point response to every comment as follows.

(1) In table 1, air temperature is monitored at two heights (2m and 4m). In Figure 2, the air temperature sensor (the arrow indicated) looks like at 4m height, if the GST sensor is rightly at 2m. In Figure 3a, which one was shown here?

**Response**:

The air temperature in Table 1 is right. In Fig.2, there are two air temperature sensors that installed in 2m and 4m height, but only the sensor at 4m height is labelled with red arrow.

The LST sensor at 2m height is also right, because it is non-contact infrared radiation sensor. In the Fig. 3a, air temperature at 2m height is shown, we have added this information in the Line of 164.

To be clear, we explained the information in the figure legend of Fig. 3 as follows:

**Figure 3.** Time series of meteorological variables from 2010 to 2020 at Mahan Mountain, including air temperature at 2 m height (a), land surface temperature (b), precipitation at 1.6 m height (c), shortwave radiation at 2 m height (d), longwave radiation at 2 m height (e), water vapor pressure at 4 m height (f), relative humidity at 4 m height (g), wind speed & direction at 2 m height (h). The temporal resolution of precipitation data is daily scale, and hourly scale for other all variables.

(2) The authors should note the date of the field picture (during the station was established?). If the surface always looks like this, the observations should be significantly different from the surrounding. Because the vegetation looks different from the surrounding area.

**Response**:

Fig. 2a was recorded on June 15, 2020, which has been added in the figure caption of Fig. 2a.

When the station was established, there was no fence around the whole field. Subsequently, we found that some sensors were damaged by yaks and sheep. Therefore, we had to setup the fence to protect these sensors from local grazing herds. However, the existence of no fence can cause the significant differences in vegetation condition inside and outside of the fence. We mowed vegetation periodically inside of the fence and try to keep same condition of inside and outside. We explained this in the figure legend as follows:

Note that we selected flat ground with the same vegetation type to set up the instruments. While some instruments were destroyed by animals, so we set up a fence to protect the instruments. There were slight differences in the vegetation biomass during the following years. (Line 157-160)

(3) Figure 3, what temporal resolution is showing here? Hourly, daily, or others?

**Response**:

Time series of meteorological variables in Fig. 3 are based on daily precipitation data (c) and hourly records for all other variables (a–b, d–h). We have added this information in the revised manuscript, which is as follows:

The temporal resolution of precipitation data is daily scale, and hourly scale for other all variables. (Line 175-176)

(4) For precipitation, I don't know what mean annual precipitation is (line 186). From Figure 3c, the maximum is < 30mm. I guess they are annual totals?

**Response**:

Thank you very much for pointing this out. The "mean annual precipitation (line 186)" has been changed into "annual total precipitation" in the revised manuscript, the detailed descriptions are as follows:

The observed local annual total precipitation was 318.6 ± 54.3 mm from 2010 to 2020, and the minimum and maximum annual total precipitation occurred in 2015 and 2018 with values of 258.3 mm and 443.9 mm, respectively.

(5) Figure 6, y-axis: the unit is meter? Or cm? Why was there an anomaly peak at ~60 (m/cm?) at 9# during Aug 2016 (i.e., Figure 6e)?

**Response**:

The unit of depth in y-axis is centimeter, in the revised manuscript, we have changed it.

There exist an anomaly peak value of soil moisture at 9# during Aug. 2016, these data are acquired using standard oven drying method. We analyzed it and concluded that there might exist subsurface flow at depth of 60cm or so, which could cause greater soil moisture values. In the revised manuscript, we have added this information, which is as follows:

The abnormally high value at a depth of 60 cm at site 9# during August 2016 is likely related to the existence of subsurface flow. (Line 359-360)

(6) Table 2, range of VWC is 0-100%? Generally, it should be 0-1 because VWC has a unit of m3/m3. Actually, the author did show that in Figure 5.

**Response**:

The range of VWC should be 0-1, in the revised manuscript, we have changed it in Table 2. We also checked the unit of VWC in Figure 5, and found it is right.

(7) Figure 7, was this borehole drilled in 2010? Line 314, the authors stated that the borehole was drilled in Aug 2008. If the borehole was drilled in 2010, the ground temperatures measured in 2010 are not reasonable because of the disturbance of drilling. I also worry about the disturbance of drilling on the records during a couple of years in the beginning. The active layer might return to normal after one year. But the boreholes may not be.

**Response**:

Thanks for the comment. The borehole was drilled in August 2008, and then in mid-December 2008, the temperature probe was placed at a predetermined depth and monitored for the first time. From May 2009, permafrost temperature data for each half-hour was automatically recorded by a data logger. Therefore, the ground temperatures measured in 2010 are reasonable because it has been more than a year after the borehole was drilled.

To be clear, we added the installation time of thermistor probes in Permafrost temperature section as follows:

In mid-December 2008, twenty thermistors were installed at different depths in the borehole (0.1 m, 0.4 m, 0.9 m, 1.4 m, 1.9 m, 2.4 m, 3.4 m, 6.4 m, 7.4 m, 9.4 m, 11.4 m, 13.4 m, 15.4 m, 17.4 m, 19.4 m, 21.4 m, 23.4 m, 25.4 m, 27.4 m, and 28.4 m). (Line 367-370)

(8) Figure 7, the horizontal dashed line is 1.2m depth? It significantly departures from the scale (because it looks like at the middle of 0-5m).

**Response**:

We appreciate your careful review, we have changed the tick mark of the horizontal dashed line into "2 m", which is as follows:

[Figure]

**Figure 9.** Ground temperature in the permafrost borehole drilled in 2010 at Mahan Mountain.

(9) Are the thermistors in the borehole shifting during the monitoring period? Are they calibrated every year or at a constant frequency in lab? Modern sensors and transmitters are electronic devices, and the reference voltage, or signal, may drift over time due to temperature, pressure, or change in ambient conditions.

**Response**:

  To prevent the thermistors in the borehole shifting during the monitoring period, we setup a steel wire running through the borehole, and the cable wrapped with thermistors is fixed to the steel wire, which can ensure the cable is vertical and prevent the thermistors moving in the borehole.

  In general, we calibrate these thermistors every year in the State Key Laboratory of Frozen Soil Engineering, Chinese Academy of Sciences. During the calibration process, another cable wrapped with similar thermistors is setup to guarantee data continuity. However, due to the sensor failure in 2012, then we manually measured the permafrost temperature using the cable wrapped with similar thermistors, these thermistors were also calibrated periodically to

guarantee enough precision. To be clear, this information is added in the Data
description section. (Line 134-144)

(10) Are the depths of all sensors in soil calibrated every year? Because thaw subsidence
and frost heaven may have significant influences on the monitoring depths of soil sensors.
In other words, after a couple of years, the sensors probably depart significantly from the
initial depths.

**Response:**

Indeed, the thaw subsidence and frost heaven can exert significant
influence on the monitoring depths of thermistors. As mentioned above, we have
setup a steel wire running through the borehole, and the cable wrapped with
thermistors is fixed to the steel wire, which can ensure the cable is vertical and
prevent the thermistors shifting during the monitoring period. In fact, except the
shallow or near surface soil temperature shows sharply variations, the ground
temperature in other depths showed slight changes. For the sensors in the active
layers, we cannot calibrate the depths of all sensors every year. Thanks for the
suggestion. To be clear, we explained this in the methods section as follows:

However, for the sensors in the active layers, we cannot calibrate the depths
of all sensors every year, which may lead to some errors. (Line 141-143)

2 A confusing point is that long-term MAAT is -1.4 (Line 120) while long-term MAGST is
also -1.4 C (Line 171). As I know, generally, MAGST is higher than MAAT (including sites
on the Qinghai-Tibetan Plateau, Alaska, and other permafrost sites). But at this site, they
look equally. The authors have to explore and explain this feature.

**Response:**

Thank you very much for pointing this out. We are sorry for the mistake that
the "GST" should be replaced by "LST". The measured land surface temperature
is the radiative skin temperature by the IRP-P at a height of 2 m. The surface
could be snow, grass, and a mixture of them. Therefore, the variable should be

"LST". We didn't observe he ground surface temperature, the top depth of recorded soil temperature for active layer is 10 cm. We calculated local MAAT (2m) and MALST during observation period, which is -1.35°C and -1.43°C, respectively, so they are closely consistent. In the revised manuscript, we improved related descriptions.

3 The authors should calculate and explain the thermal offset, i.e., the difference between mean annual soil temperature at the bottom of the active layer and mean annual soil surface (~0cm) temperature (Romanovsky and Osterkamp, 1995). If the values in the manuscript are true, GST is -1.4C and MAGT at 2.4m is ~-0.1C. The offset would be roughly +1.3 C. The authors have to explain such a large reversed thermal offset (because it generally is negative. Of cause, some people (Lin et al., 2015; Luo et al., 2018) 'found' small positive offsets).

**Response**:

As stated in the above question, the value of MAGST is larger than +0.97°C (MAST at -10 cm). Therefore, the thermal offset = TTOP - MAGST= -0.1°C - (> +0.97°C) > -1.07°C (negative). The result is consistent with the general understanding for thermal offset in the permafrost regions. In the revised manuscript, we added related descriptions as follow:

Under the influence of the freeze–thaw process, the thermal state of the active layer is not constant during the whole year. In addition, the difference in thermal conductivity between the frozen and thawed ground causes a "negative thermal offset", which is defined as the difference between the mean annual soil temperature at the bottom of the active layer (TTOP) and the mean annual soil surface (~0 cm) temperature (MAGST) (Burn and Smith, 1988). In this study, the value of MAGST is larger than +0.97 °C (MAST at 10 cm). Therefore, the thermal offset = TTOP - MAGST= -0.1 °C - (> +0.97 °C) > -1.07 °C. This result is consistent with the general understanding of thermal offset in the permafrost regions (Romanovsky and Osterkamp, 1995). (Line 272-280)

**References:**

Burn, C. R., & Smith, C. A. S.: Observations of the" thermal offset" in near-surface mean annual ground temperatures at several sites near Mayo, Yukon Territory, Canada, Arctic, 99-104, https://www.jstor.org/stable/40510685, 1988.

Romanovsky, V. E., & Osterkamp, T. E.: Interannual variations of the thermal regime of the active layer and near‑surface permafrost in northern Alaska. Permafrost and Periglacial Processes, 6(4), 313-335, https://doi.org/10.1002/ppp.3430060404, 1995.

4 References should be appropriate rather than load the manuscript with fancy references (same with the comment from Reviewer 1, however, I would indicate more specifically). An example is Line 39. Boike et al. (2019) only described a site-level dataset (like this manuscript) and gave not any hemispheric summary. Meanwhile, the authors should use terms (permafrost and permafrost region) correctly. Zhang et al. (1999) and Zhang et al. (2000) indicated that permafrost region underlies approximately $22.79 \times 10^6$ km$^2$ or 23.9% of the exposed land area of the Northern Hemisphere. Certainly, that permafrost may not be present everywhere within permafrost region. They present estimates that indicate that the actual area underlain by permafrost is smaller, ranging approximately from 12.21 to $16.98 \times 10^6$ km$^2$ (the source of 15 million km$^2$). Whatever, I don't know how only 15 million km2 can be 24% of the land surface area.

**Response:**

Thanks for pointing this out. In the revised version, we carefully checked all the references and made necessary changes. We also changed the sentences in the revised manuscript, which is as follows:

As a major component of the cryosphere, the area underlain by permafrost ranges from $12.21 \times 10^6$ km$^2$ to $16.98 \times 10^6$ km$^2$, or from 12.8% to 17.8% of the terrestrial landscape in the Northern Hemisphere (Zhang et al., 2000). (Line 34-37)

5 Finally, I totally agree with the comments from Reviewers 1 and 2. The language should be re-edited by English-native-speakers, although it is understandable but need readers to take heavy efforts.

**Response:**

Thanks for the suggestion. We polished our language by American Journal Experts (https://www.aje.com/) which is a partner of many publishing groups. The changes were highlighted in blue so that they may be easily identified. The editing certificate by AJE were presented as follows:

[Figure]

---

## Author Response (AR2)

Dear Editor,

We are especially grateful to your helpful comments. According to those suggestions, we have made thorough revisions to the manuscript. In particular, we modified related grammatical errors or problematic expressions. The changes made to the text are highlighted in blue so that they may be easily identified in the revised manuscript.

Yours sincerely,

Tonghua Wu on behalf of all co-authors

**Response to comments:**

1) The title seems lengthy and tedious, particularly the description of the location. There is also missing a comma before "and". I would encourage the authors to simplify it a bit to be clearer.

**Response**:

Thanks for the suggestion. In the revised manuscript, we changed the title to "Permafrost, active layer, and meteorological data (2010–2020) at the Mahan Mountain relict permafrost site of Northeastern Qinghai-Tibet Plateau".

2) It is common that the authors had miss used "the", both missing and wrongly used when there were not needed. For example:

a. Line 21, missing "the" before "Mahan". The similar issue can also be found in other parts of the manuscript.

b. Line 116, missing "the" before "relict" unless the authors believe that no relict permafrost in the world is sensitive to climate warming.

c. Line 188, missing "the" before "Mahan".

**Response**:

In the revised manuscript, we added "the" before "Mahan" and "relict", and all of similar issues are also improved.

3) Line 55, 273, missing comma before "and"; line 408, missing comma before "north" and "central".

**Response**:

Changed.

4) The second last paragraph in the Introduction also dedicated to describing the site in the Mahan Mountain, so as the section 2.1. I would suggest the authors to reorganize the two paragraphs to make the narration clearer.

**Response**:

Thanks for the suggestion. In the revised manuscript, we improved it. In the second last paragraph of Introduction, we mainly described the location and particularity of the Mahan Mountain relict permafrost (Line 74-87). In the section 2.1, we described establishment, early field work, and main features of the Mahan Mountain relict permafrost observation site (Line 97-112).

5) Any citations to the statements in line 110 and 126?

**Response**:

Thanks for the suggestion. In Line 110, the description of "The original permafrost area was approximately 0.16 km$^2$, the area of which has recently been reduced to 0.13 km$^2$" was cited from the literature of Xie et al. (2013), in the revised manuscript, we added this citation (Line 105).

In Line 126, the results of mean annual air temperature and the duration of negative air temperature are acquired from the observed meteorological data of the Mahan Mountain monitoring site, and this sentence has explained clearly (Line 120-123).

Xie, C., Gough, W. A., Tam, A., Zhao, L., and Wu, T.: Characteristics and Persistence of Relict High-Altitude Permafrost on Mahan Mountain, Loess Plateau, China, Permafr. Periglac. Process., 24, 200-209, https://doi.org/10.1002/ppp.1776, 2013.

6) Figure 1, the first sentence is not grammatically correct.

**Response**:

We modified it as:

The location of Mahan Mountain relict permafrost region in China (a), the spatial distribution of permafrost and monitoring sites in the study region (b). (Line 114-115)

7) Line 121, "derived based on …" is not a correct expression.

**Response**:

We modified it as:

The permafrost distribution of the Mahan Mountain is derived from Xie et al. (2013). (Line 117-118)

8) Line 122, "map" and "data" were redundant, suggest drop one of them.

**Response**:

We deleted "map" in the revised manuscript.

9) Line 159, change "the same …" to "homogeneous"?

**Response**:

Changed.

10) Line 160, change "While" to "After"?

**Response**:

Changed.

11) Line 170, change "is as follows" to "is presented as following (Table 1)".

**Response**:

Changed.

12) Line 210, change "but" to "and".

**Response**:

Changed.

13) Line 227, change "0-100%" to "100%"?

> **Response**:
>
> Changed.

14) I think it would be helpful to clarify that 6999 was adopted as the fill value.

> **Response**:
>
> Thanks. We clarified this in the revised version as:
>
> (1) The negative values for wind directions were replaced by 6999 (Line 232-233).
>
> (2) In the case of missing data persisting for a longer time, we filled them with 6999. Overall, all the missing or abnormal soil temperature and soil volumetric water content data were replaced with 6999 (Line 261-263).
>
> (3) Quality control was carried out to check whether the data were missing or invalid, which was replaced by 6999 as no data (Line 378-380).

15) Line 246, 247, .., please remove "data" because data was not the measuring subjects.

> **Response**:
>
> Thanks for the suggestion. In the revised manuscript, we removed the "data".

16) Line 250, I would suggest the authors to clarify the resampling method.

> **Response**:
>
> We modified the sentence to "We finally resampled the 30-minute soil temperature and soil volumetric water content data into daily data by averaging the half-hourly data within a day." (Line 245-247)

17) Table 3 and other parts of the manuscript. It should be expresses as "#1" instead of "1#".

> **Response**:
>
> In the revised manuscript, we changed all similar expression both in text

and Figures.

18) Line 370, it is incorrect to express "at … mountain". Please use "in/on" or clarify it to be the site.

**Response**:

Thanks for the suggestion. In the revised manuscript, we changed it to "at the Mahan Mountain permafrost region".

19) Line 407/409, change "/a" to "/year" or the other way around to be consistent through the manuscript.

**Response**:

Changed.

20) Fig. 4, Fig. 5, and Fig. 10: Please add x-axis title to make them consistent with other figures in the manuscript.

**Response**:

Thanks for the suggestion. In the revised manuscript, we added x-axis title in related Figures.

21) Line 409-411, the expression "Russian European north, western and central Siberia, …" seems confusing.

**Response**:

This expression was cited from the Smith et al. (2022), which was as follows:

Table 1 | Average regional rates of ALT changes from CALM network data

| Region | Average ALT change (cm per year)[a] | Range of ALT change (cm per year) | Number of sites[b] |
|---|---|---|---|
| Alaska North Slope | 0.2 | −0.1–0.5 | 25 |
| Alaska interior | 0.9 | 0.2–2.7 | 5 |
| Canada (Mackenzie Valley and eastern Arctic | 0.0 | −1.0–0.7 | 7 |
| Nordic (including Svalbard and Greenland)[c] | 1.3 | 0.5–3.8 | 7 |
| Russian European north, western and central Siberia | 1.3 | −0.1–3.7 | 20 |
| Northeastern Siberia (including Chuktoka and Kamchatka) | 0.5 | −0.5–1.9 | 24 |
| Swiss Alps[d] | 10.5 | −1.8–31.6 | 9 |
| Antarctica | 0.1 | −1.5–2.5 | 12 |

Smith, S. L., O'Neill, H. B., Isaksen, K., Noetzli, J., and Romanovsky, V. E.: The

changing thermal state of permafrost. Nat Rev Earth Environ., 3(1): 10–23. https://doi.org/10.1038/s43017-021-00240-1, 2022.

In the revised manuscript, we modified it to "northern Russian European, western and central Siberia".

22) Both terms of "Qinghai-Tibet Plateau" and "Tibetan Plateau" occurred in the manuscript, I wonder if the authors think they could be modified to be consistent?

**Response**:

Thanks for the suggestion. In the revised manuscript, we changed "Tibetan Plateau" to "Qinghai-Tibet Plateau".

Furthermore, we read through the context of the manuscript. We added three important references in the context. At the same time, we updated the reference list of the manuscript.

Line 49 and Line 608: Wang, D., Wu, T., Zhao, L., Mu, C., Li, R., Wei, X., Hu, G., Zou, D., Zhu, X., Chen, J., Hao, J., Ni, J., Li, X., Ma, W., Wen, A., Shang, C., La, Y., Ma, X., and Wu, X.: A 1 km resolution soil organic carbon dataset for frozen ground in the Third Pole, Earth Syst. Sci. Data, 13, 3453-3465, https://doi.org/10.5194/essd-13-3453-2021, 2021.

Line 49 and Line 626: Zhang, G., Ran, Y., Wan, W., Luo, W., Chen, W., Xu, F., and Li, X.: 100 years of lake evolution over the Qinghai-Tibet Plateau, Earth Syst. Sci. Data, 13, 3951-3966, https://doi.org/10.5194/essd-13-3951-2021, 2021.

Line 50 and Line 588: Shogren, A. J., Zarnetske, J. P., Abbott, B. W., Bratsman, S., Brown, B., Carey, M. P., Fulweber, R., Greaves, H. E., Haines, E., Iannucci, F., Koch, J. C., Medvedeff, A., O'Donnell, J. A., Patch, L., Poulin, B. A., Williamson, T. J., and Bowden, W. B.: Multi-year, spatially extensive, watershed-scale synoptic stream chemistry and water quality conditions for six

permafrost-underlain Arctic watersheds, Earth Syst. Sci. Data, 14, 95-116, https://doi.org/10.5194/essd-14-95-2022, 2022.